# Independent Stage Classification for Gastroesophageal Junction Adenocarcinoma

**DOI:** 10.3390/cancers15215137

**Published:** 2023-10-25

**Authors:** Yuki Hirata, Yi-Ju Chiang, Jeannelyn S. Estrella, Prajnan Das, Bruce D. Minsky, Mariela Blum Murphy, Jaffer A. Ajani, Paul Mansfield, Brian D. Badgwell, Naruhiko Ikoma

**Affiliations:** 1Department of Surgical Oncology, The University of Texas MD Anderson Cancer Center, Houston, TX 77030, USA; yhirata@mdanderson.org (Y.H.); ychiang1@mdanderson.org (Y.-J.C.); pmansfie@mdanderson.org (P.M.); bbadgwell@mdanderson.org (B.D.B.); 2Department of Pathology, The University of Texas MD Anderson Cancer Center, Houston, TX 77030, USA; jsestrella@mdanderson.org; 3Department of Radiation Oncology, The University of Texas MD Anderson Cancer Center, Houston, TX 77030, USA; prajdas@mdanderson.org (P.D.); bminsky@mdanderson.org (B.D.M.); 4Department of Gastrointestinal Medical Oncology, The University of Texas MD Anderson Cancer Center, Houston, TX 77030, USA; mblum1@mdanderson.org (M.B.M.); jajani@mdanderson.org (J.A.A.)

**Keywords:** gastroesophageal junction adenocarcinoma, TNM staging, preoperative chemotherapy, preoperative chemoradiotherapy, National Cancer Database

## Abstract

**Simple Summary:**

In gastroesophageal junction (GEJ) adenocarcinoma cases, a prognosis based on ypTNM staging could be affected by preoperative therapy. In this study, we investigated 11,340 patients with GEJ adenocarcinoma who received preoperative therapy followed by curative-intent surgery and examined the clinicopathologic factors associated with overall survival (OS). We found that median OS durations in patients with GEJ adenocarcinoma, when sorted by stage, were substantially different from those of patients with adenocarcinoma of the upper or middle esophagus or of the non-cardia stomach. Additionally, preoperative chemoradiotherapy (CXRT) was associated with lower OS rates than chemotherapy after adjustment for the ypT and ypN categories. These results indicate that we should develop an independent GEJ staging system, rather than separating GEJ cancers into either esophageal or gastric cancers and using those staging systems, for better OS prediction. The OS prediction nomogram developed in this study, which included a preoperative therapy regimen, provided reasonable OS prediction.

**Abstract:**

In gastroesophageal junction (GEJ) adenocarcinoma cases, a prognosis based on ypTNM staging could be affected by preoperative therapy. Patients with esophageal adenocarcinoma and gastric adenocarcinoma who underwent preoperative therapy followed by surgical resection from 2006 through 2017 were identified in the National Cancer Database. To enable stage-by-stage OS comparisons, tumors were classified into four gross ypTNM groups: ypT1/2, N-negative; ypT1/2, N-positive; ypT3/4, N-negative; and ypT3/4, N-positive. Prognostic factors were examined, and an OS prediction nomogram was developed for patients with abdominal/lower esophageal and gastric cardia adenocarcinoma, representing GEJ cancers. We examined 25,463 patient records. When compared by gross ypTNM group, the abdominal/lower esophageal and gastric cardia adenocarcinoma groups had similar OS rates, differing from those of other esophageal or gastric cancers. Cox regression analysis of patients with GEJ cancers showed that preoperative chemoradiotherapy was associated with shorter OS than preoperative chemotherapy after adjustment for the ypTNM group (hazard ratio 1.31, 95% CI 1.24–1.39, *p* < 0.001), likely owing to downstaging effects. The nomogram had a concordance index of 0.833 and a time-dependent area under the curve of 0.669. OS prediction in GEJ adenocarcinoma cases should include preoperative therapy regimens. Our OS prediction nomogram provided reasonable OS prediction for patients with GEJ adenocarcinoma, and future validation is needed.

## 1. Introduction

The incidence of gastroesophageal junction (GEJ) adenocarcinoma has been increasing in the past 30 years and continues to be a significant public health problem worldwide [1,2,3,4]. Because of the cancer’s location between the esophagus and the stomach, optimal classification, staging, and treatment strategies specifically for GEJ adenocarcinoma are still controversial. In the latest (eighth) edition of the American Joint Committee on Cancer (AJCC) staging manual [5,6], GEJ adenocarcinomas are staged using one of two staging systems based on the location of the epicenter of the tumor. Tumors with epicenters located more than 2 cm below the GEJ into the proximal stomach (i.e., Siewert type III GEJ cancers) are staged as gastric cancers, whereas those with epicenters located more than 2 cm above or within 2 cm of the GEJ (i.e., Siewert type I and type II cancers, respectively) are staged as esophageal cancers. The use of different staging systems based only on the location of the tumor’s epicenter does not effectively guide medical and surgical strategies, while it causes misclassification and may lead to inaccurate prognostic predictions [7,8].

Multiple randomized controlled trials have reported superior survival in patients with esophageal and gastric adenocarcinoma who received perioperative or preoperative therapy rather than surgery alone, and use of preoperative therapy has sharply increased over the past decade [3]. However, the optimal regimens and durations of preoperative therapy, particularly for GEJ adenocarcinoma, are still unknown [9,10,11]. Because of the increasing use of preoperative therapy, the eighth edition of the AJCC staging manual included post-preoperative therapy pathologic staging categories (ypTNM) for patients with esophageal and gastric cancer who received any therapy before surgery [12]. Preoperative therapy often leads to pathologic downstaging, and the degree of pathologic downstaging varies according to the preoperative therapy regimen and treatment duration, particularly if radiotherapy is included [13,14]. Whether survival predictions based on ypTNM classifications are affected by preoperative therapy regimens is unknown. 

To address this gap in knowledge, we conducted the current study to determine whether TNM classification has survival implications in GEJ adenocarcinoma (i.e., whether we need to continue to use separate staging systems for GEJ adenocarcinomas based on tumor location) and whether survival implications based on ypTNM categories are similar after preoperative chemoradiotherapy (CXRT) or chemotherapy for GEJ adenocarcinomas (i.e., whether we can use the same ypTNM staging system regardless of the preoperative therapy regimen used). We used the National Cancer Database (NCDB) to investigate prognostic factors in patients with GEJ adenocarcinoma who received preoperative therapy followed by curative-intent surgery and examined the clinicopathologic factors associated with overall survival (OS). 

## 2. Materials and Methods

### 2.1. Data Source

For our analysis, we examined the NCDB records of patients with cancer of the esophagus and stomach from 2006 to 2017. The NCDB, an oncology database jointly supported by the American Cancer Society and the American College of Surgeons, includes registry data collected from more than 1500 facilities accredited by the Commission on Cancer; these records represent more than 70% of new cancer cases in the United States [15]. The NCDB is a publicly available, deidentified dataset with strict adherence to Health Insurance Portability and Accountability Act regulations. Analysis from this dataset was exempt from full review by the Institutional Review Board at The University of Texas MD Anderson Cancer Center.

We queried NCDB for all patients aged 18 years and older whose records included International Classification of Disease of Oncology, Third Edition (ICD-O-3) topography codes C152-C155 (C152, abdominal esophagus; C153, upper third of the esophagus; C154, middle third of the esophagus; C155, lower third of the esophagus) and C160-C166 (gastric cardia, gastric fundus, gastric body, gastric antrum, gastric pylorus, gastric lesser curvature, and gastric greater curvature). In addition, we searched for ICD-O-3 histology codes 8140 (adenocarcinoma, not otherwise specified), 8144 (adenocarcinoma, intestinal type), 8145 (adenocarcinoma, diffuse type), and 8490 (signet ring cell adenocarcinoma) [3,11]. We defined patients with GEJ adenocarcinoma as those who had adenocarcinoma of the lower third of the esophagus, abdominal esophagus, or gastric cardia. 

### 2.2. Patient Selection

Our analysis included all patients aged 18 years and older who had undergone preoperative therapy followed by surgical resection (esophagectomy or gastrectomy) for nonmetastatic invasive esophageal or gastric adenocarcinoma (Figure 1). To improve the validity of the analysis, we did not include patients for whom the time between the date of chemotherapy or CXRT initiation and the date of surgery was fewer than 30 days or longer than 365 days [14,16]. 

### 2.3. Patient, Tumor, and Treatment Characteristics

Patient demographic, clinical, and tumor characteristics were collected, including sex; age, and race/ethnicity of the patient; Charlson–Deyo score; primary tumor location; histologic grade; tumor size; number of days between postoperative therapy and surgery; surgery type; and clinical T, clinical N, ypT, and ypN categories. The T and N categories were defined according to the eighth edition of the AJCC staging manual. The collected treatment information included the type of preoperative therapy, margin status, and postoperative mortality. 

### 2.4. Statistical Analysis

The primary outcome of the study was OS, which was defined as the time from diagnosis to death. The main variables in the study were tumor location (upper or middle esophagus, lower or abdominal esophagus, gastric cardia, or other stomach) and preoperative therapy regimen (CXRT or chemotherapy only). To enable stage-by-stage comparisons, we classified tumors into one of four gross ypTNM groups: ypT1/2, N-negative; ypT1/2, N-positive; ypT3/4, N-negative; and ypT3/4, N-positive. OS was compared according to tumor location using the Kaplan–Meier method and the log-rank test. Baseline differences between patients were examined using the chi-square test for categorical variables and a two-sided Student’s *t* test or Mann–Whitney U test for continuous variables. As mentioned above, GEJ adenocarcinoma was defined as adenocarcinoma of the lower third of the esophagus, abdominal esophagus, or gastric cardia, and the prognostic factors for patients with these cancers were further analyzed. A Cox proportional hazards regression model was used to identify the significant variables affecting OS. Stepwise model selection was implemented until all remaining predictors had *p* values less than 0.05, and the hazard ratio (HR) and 95% confidence interval (CI) were calculated. Then, using the data from the final multivariable model, we created a nomogram to predict the probability of 3-year and 5-year OS. *p* values less than 0.05 were considered statistically significant. The concordance index and time-dependent area under the curve were calculated by bootstrapping to evaluate the discriminative ability. Statistical analysis was performed using SAS Enterprise Guide 8.3 (SAS Institute, Cary, NC, USA) and STATA version 14.1 (StataCorp LLC, College Station, TX, USA). The nomogram was drawn using R version 4.1.2 (R Foundation for Statistical Computing, https://www.r-project.org/).

## 3. Results

### 3.1. Patient Characteristics

Data from 25,463 patients who had undergone preoperative therapy for resectable esophageal and gastric adenocarcinoma were used for the initial analysis (Figure 1). Demographic, clinical, tumor, and treatment characteristics are summarized in Table 1 according to tumor location. Overall, the mean age was 63 years, 82.1% of patients were men, and 85.0% were White. The most frequent clinical T category was T3 (62.9%), and the most frequent clinical N category was positive (56.6%). Most patients (73.4%) underwent preoperative CXRT, and the proportion of patients who received CXRT was higher among patients with esophageal adenocarcinoma (91.8% for upper or middle esophageal adenocarcinoma and 91.4% for lower or abdominal esophageal adenocarcinoma). Radiotherapy was used less frequently among patients with non-cardia gastric adenocarcinoma; 78.7% of patients with gastric cardia adenocarcinoma received CXRT, and 8.1% of those with other stomach adenocarcinomas received CXRT.

### 3.2. Five-Year and Three-Year OS Rates by Gross ypTNM Group

Survival outcomes were available for all 25,463 patients in the study. With a median follow-up time of 2.6 years, the median OS duration was 3.7 years, and the 5-year OS rate was 43%. Table 2 shows the 3-year OS rate, 5-year OS rate, and median OS duration by ypTNM group and tumor location (Kaplan-Meier curves are shown in Appendix A). The median OS durations were similar between patients with lower or abdominal esophageal adenocarcinomas and those with gastric cardia adenocarcinomas within the gross ypTNM groups. For example, in the ypT3/4, N-negative group, the median OS was 3.3 years for those with lower or abdominal esophageal adenocarcinoma and 3.9 years for those with gastric cardia adenocarcinoma. In contrast, the median OS duration in the ypT3/4, N-negative group was substantially shorter among those with upper or middle esophageal adenocarcinoma (2.1 years) and longer among those with other stomach adenocarcinoma (7.1 years). These results indicate that it is reasonable to combine patients with lower or abdominal esophageal adenocarcinoma and those with gastric cardia adenocarcinoma into a single GEJ category for post-preoperative therapy staging, and this staging system should be separate from that used for patients with other esophageal or gastric adenocarcinoma.

### 3.3. Prognostic Factors for GEJ Adenocarcinoma

We included patients with lower or abdominal esophageal adenocarcinoma and those with gastric cardia adenocarcinoma in our analysis of the prognostic factors for GEJ adenocarcinoma. The factors associated with OS in GEJ adenocarcinoma, which we identified with a Cox proportional hazards regression model, are shown in Table 3. Tumor location (lower or abdominal esophagus), older age, male sex, poorly differentiated histologic grade, positive margins, and higher ypT and ypN categories were associated with poor prognosis (all *p* < 0.001). Most notably, preoperative CXRT was associated with a worse OS compared with chemotherapy (HR 1.31, 95% CI 1.24–1.39, *p* < 0.001), likely owing to the paradoxical effect of more frequent downstaging after CXRT, and patients with gastric cardia adenocarcinoma had slightly better OS compared with those with lower or abdominal esophageal adenocarcinoma (HR 0.91, 95% CI 0.87–0.95, *p* < 0.001), after adjustment for ypT and ypN category. The effect of the treatment regimen and tumor location on median OS duration appeared similar across different ypTNM groups, justifying the creation of a combined nomogram for survival prediction.

### 3.4. Development of the Competing-Risk Nomogram Predicting OS

The competing-risk nomogram predicting the 3- and 5-year OS rates in patients with GEJ adenocarcinoma was established based on a selection of prognostic factors (Figure 2). Each variable subtype was assigned a score. A straight line to determine the estimated probability of survival can be drawn at each time point on the total point scale, according to the total points. The nomogram showed that the ypN2 and ypN3 categories, followed by the ypT4 category, were the most important factors determining prognosis. Positive resection margins, age ≥ 70 years, and the use of CXRT were equal predictors of poor OS. The concordance index was 0.833, and the bootstrapped 95% CI was 0.818–0.847. The integrated, time-dependent area under the curve was 0.669, and the 95% CI was 0.659–0.679 (Figure 3). These results indicate that this model fits well to predict OS outcome, and it was accurate, even though the first-year area under the curve was lower than 0.65. 

## 4. Discussion

Our results illustrate the limitations of using two different TNM staging systems based on tumor location to predict prognosis in GEJ adenocarcinoma. First, we found that median OS durations in patients with GEJ adenocarcinoma (lower or abdominal esophageal adenocarcinoma and gastric cardia adenocarcinoma), when sorted by stage, are substantially different from those of patients with adenocarcinoma of the upper or middle esophagus or of the non-cardia stomach. Median OS durations were also similar between patients with lower or abdominal esophageal adenocarcinoma and those with gastric cardia adenocarcinoma. These results indicated that we should develop an independent GEJ staging system, rather than separating GEJ cancers into either esophageal or gastric cancers and using those staging systems, to achieve better OS prediction. Second, preoperative CXRT was associated with lower 5-year OS rates compared with chemotherapy, after adjustment for ypT and ypN categories, likely owing to the paradoxical effect of more frequent downstaging after CXRT. Therefore, even within the same ypT and ypN categories, OS differs by preoperative treatment regimen used, and a single ypTNM staging system would not provide accurate OS prediction after preoperative therapy and surgery. These findings suggest that a novel OS prediction system for GEJ adenocarcinoma that considers various clinicopathologic factors, including the type of preoperative therapy, is needed. The OS prediction nomogram developed in the current study, which included the preoperative therapy regimen (CXRT or chemotherapy alone), provided reasonable OS prediction for patients with GEJ adenocarcinoma who underwent preoperative therapy and curative-intent surgery. Future validation is needed.

Multidisciplinary treatment regimens for esophageal and gastric cancers have evolved over the past two decades; however, there is a paucity of data specific to GEJ adenocarcinomas. The current recommendations for preoperative therapy regimens were made based on studies that included not only GEJ tumors, but also either esophageal or gastric cancers. The MAGIC and FLOT4 trials, for example, have shown the benefits of preoperative chemotherapy for various gastroesophageal tumors, including GEJ adenocarcinoma [9,10,17]. The MAGIC trial, conducted in patients with resectable adenocarcinoma of the stomach (74%), GEJ (11%), or lower esophagus (15%), showed that preoperative and postoperative treatment with epirubicin, cisplatin, and fluorouracil improved the 5-year OS more than surgery alone (5-year OS rate 36% compared with 23%, HR 0.66, 95% CI 0.53–0.81, *p* < 0.001). In the FLOT4 trial, in which approximately half of the patients had GEJ adenocarcinoma and the other half had gastric adenocarcinoma, preoperative and postoperative treatment with docetaxel, oxaliplatin, leucovorin, and fluorouracil improved the median OS duration more than perioperative epirubicin, cisplatin, and fluorouracil did (median OS 50 months compared with 35 months, HR 0.77, 95% CI 0.63–0.94, *p* = 0.012). In the PRODIGY trial, 5.6% of patients had GEJ adenocarcinoma. Preoperative docetaxel, oxaliplatin, and S-1 was associated with improved 3-year progression-free survival compared with upfront surgery (66.3% vs 60.2%, HR 0.70, 95% CI 0.52-0.95, *p* = 0.023).

Several randomized controlled trials have also reported a survival benefit with preoperative CXRT, including in patients with GEJ adenocarcinomas. The CROSS trial compared surgery alone with preoperative CXRT (paclitaxel, carboplatin, and 41.4 Gy radiotherapy) in patients with resectable esophageal and GEJ adenocarcinomas and squamous cell carcinomas. The median OS duration was 49.4 months in the preoperative CXRT group and 24.0 months in the surgery-alone group (HR 0.657, 95% CI 0.50–0.87, *p* = 0.003). However, only 25% of the patients in the trial had GEJ cancer, and CXRT demonstrated a greater effect in patients with squamous cell carcinoma than in those with adenocarcinoma in the subgroup analysis [18,19]. The POET study included only patients with locally advanced GEJ adenocarcinomas (clinical T3 or T4; Siewert type I–III) and compared preoperative cisplatin, leucovorin, and fluorouracil with preoperative cisplatin, leucovorin, and fluorouracil, followed by CXRT (cisplatin, etoposide, and 30 Gy radiotherapy). The 3-year OS rates were 27.7% in the preoperative chemotherapy group and 47.4% in the CXRT group (HR 0.67, 95% CI 0.41–1.07, *p* = 0.07). The results showed a favorable trend in OS in the preoperative CXRT group, although there was no statistically significant difference in either 3-year or 5-year OS rates because of the small number of enrolled patients [20,21]. Thus, the survival benefit of preoperative CXRT over chemotherapy-only regimens for GEJ adenocarcinoma remains to be determined. Ongoing randomized controlled trials are comparing perioperative CXRT and chemotherapy-only regimens in patients, including those with GEJ adenocarcinoma (TOPGEAR [22], ESOPEC [23], NEO-AEGIS [24,25], and CRITICS-II [26]), although none of these trials are specific to patients with GEJ adenocarcinomas. The collective results of these trials will provide better evidence to guide the preoperative use of CXRT over chemotherapy-only regimens for patients with GEJ adenocarcinomas.

After the recent increase in the use of preoperative therapy for both esophageal and gastric cancers [27], post-preoperative therapy staging systems were introduced. Because different types of preoperative treatment have different degrees of pathologic downstaging, patients with the same ypTNM categories may have different OS outcomes, depending on the type of preoperative treatment they received. In fact, we previously reported, using NCDB data, that preoperative CXRT was associated with a higher complete response rate in the primary tumor but not with improved OS in lower esophageal and GEJ adenocarcinoma, indicating that pathologic stage does not always correlate with OS [11]. Therefore, a novel and unique stage classification system for GEJ adenocarcinoma that considers the type of preoperative therapy was deemed necessary. In addition, because many clinicopathologic factors other than ypTNM categories significantly contribute to OS outcomes, we postulated that a nomogram could provide a more accurate prediction of OS for patients with GEJ adenocarcinoma [28,29]. The prognostic impact of each ypT and ypN category we observed suggested that each of these factors affects OS in a stepwise fashion, indicating that interaction factors between ypT and ypN categories need not be considered. This led us to develop the nomogram using ypT and ypN as independent factors. We observed a similar impact of the use of CXRT on OS in individual ypTNM groups, and thus, we considered CXRT as an independent OS prediction factor in the nomogram. The nomogram we developed in the current study used preoperative therapy regimen, tumor location, patient age, patient sex, comorbidity index, histologic grade, margin status, and ypT and ypN categories as prognostic factors, and the nomogram performed well in terms of predicting OS in patients with GEJ adenocarcinoma after preoperative therapy and surgery. More importantly, the nomogram shown in Figure 2 visually described the impact of each clinicopathologic factor on OS, such as advanced T and N categories and positive margins, in concordance with previous reports [12,16].

The current study has limitations, such as the previously described biases inherent to retrospective cohort studies [12]. In addition, limited variables with limited data quality and missing data adversely affect studies based on data in the NCDB, although the NCDB is the largest database of cancer patients in the United States and conducts serial quality checks. We had to assume that the lower esophageal, abdominal esophageal, and gastric cardia adenocarcinomas included in the NCDB were GEJ cancers because the NCDB does not provide information to indicate GEJ involvement or Siewert-type classifications. However, the current study also has strengths. First, this is the first study demonstrating the limitations of the current ypTNM staging systems for GEJ adenocarcinoma in this era of evolving preoperative therapy for GEJ adenocarcinoma. Second, we observed similar OS rates among patients with lower esophageal and gastric cardia adenocarcinoma (i.e., GEJ adenocarcinoma), and this was substantially different from OS in patients with other esophageal or gastric adenocarcinoma, justifying the creation of a unique post-neoadjuvant therapy OS prediction system for GEJ adenocarcinoma. Lastly, we developed a novel OS prediction nomogram that took into account the type of preoperative therapy and showed reasonable OS prediction, although further studies are needed for external validation.

## 5. Conclusions

The current study showed the limitations of the currently used method of predicting prognosis of GEJ adenocarcinoma using two different ypTNM staging systems (esophageal and gastric manuals), indicating a need for an independent GEJ adenocarcinoma staging system. Study results also showed that ypTNM-based OS prediction differs by preoperative therapy type (chemotherapy only or CXRT), suggesting a need for a novel OS prediction system that considers the type of preoperative therapy. The OS prediction nomogram developed in the current study, which included a preoperative therapy regimen (CXRT or chemotherapy-alone), provided reasonable OS prediction for patients with GEJ adenocarcinoma who underwent preoperative therapy and curative-intent surgery. Future validation is needed.

## Figures and Tables

**Figure 1 cancers-15-05137-f001:**
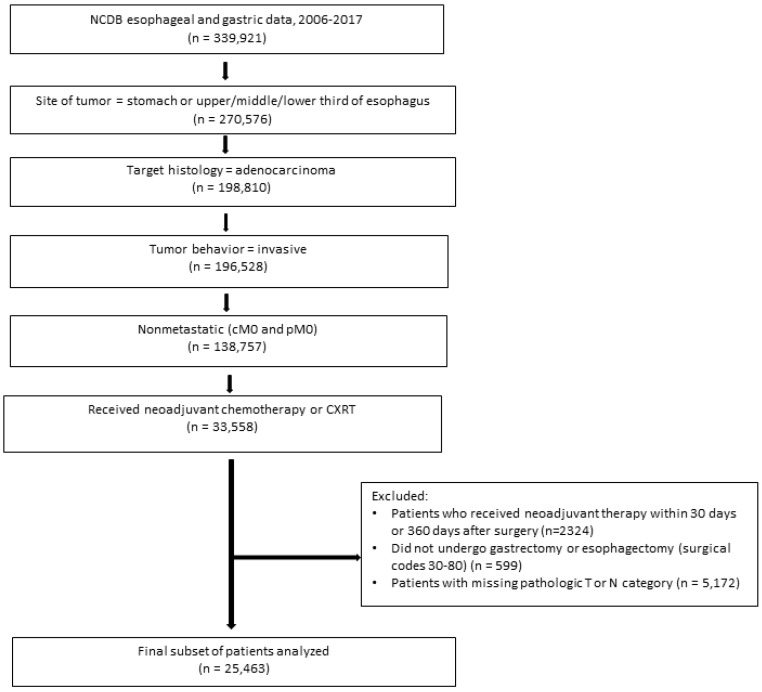
Patient selection flowchart. Abbreviations: CXRT, chemoradiotherapy; NCDB, National Cancer Database.

**Figure 2 cancers-15-05137-f002:**
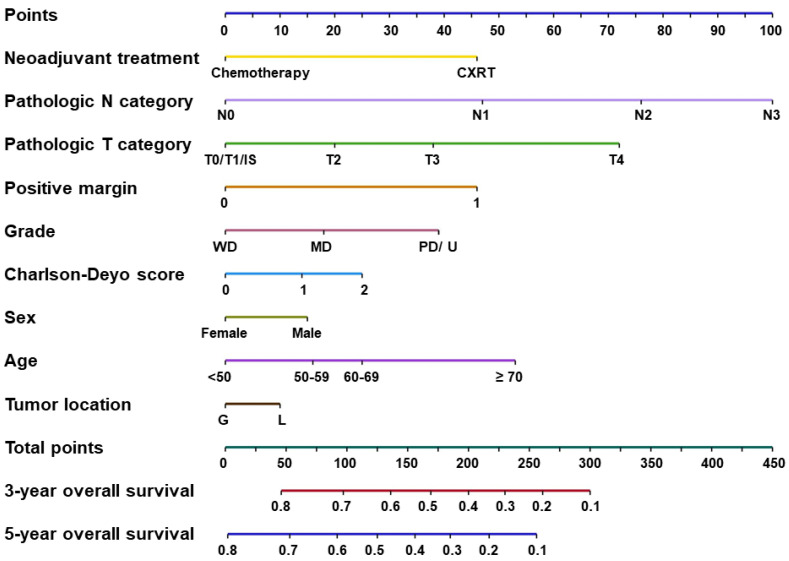
Competing-risk nomogram predicting 3- and 5-year overall survival rates in patients with gastroesophageal junction adenocarcinoma. Abbreviations: CXRT, preoperative chemoradiotherapy; IS, in situ; MD, moderately differentiated; PD/U, poorly differentiated or undifferentiated; WD, well differentiated.

**Figure 3 cancers-15-05137-f003:**
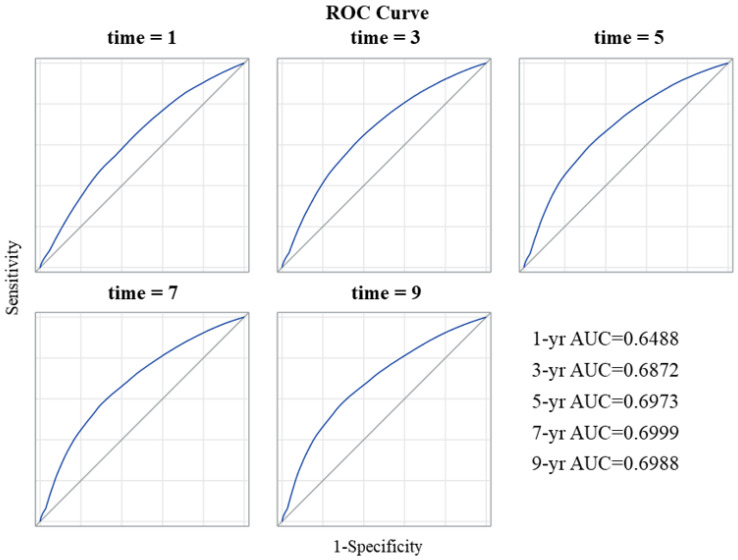
Receiver operating characteristic (ROC) curves for the nomogram. Abbreviations: AUC, area under the curve.

**Table 1 cancers-15-05137-t001:** Baseline characteristics of patients in our analysis who underwent preoperative therapy for resectable esophageal and gastric adenocarcinoma (*n* = 25,463).

	No. (%)					
		Upper/Middle Esophageal Adenocarcinoma, *n* = 645	Lower/Abdominal Esophageal Adenocarcinoma, *n* = 11,340	Gastric Cardia Adenocarcinoma, *n* = 9390	Other Stomach Adenocarcinoma, *n* = 4088	
Variable	Total, *n* = 25,463					*p*
Sex						<0.001
Male	20,904 (82.1)	551 (85.4)	10,075 (88.8)	7851 (83.6)	2427 (59.4)	
Female	4559 (17.9)	94 (14.6)	1265 (11.2)	1539 (16.4)	1661 (40.6)	
Age						0.282
≤60 years	10,001 (39.3)	242 (37.5)	4403 (38.8)	3708 (39.5)	1648 (40.3)	
>60 years	15,462 (60.7)	403 (62.5)	6937 (61.2)	5682 (60.5)	2440 (59.7)	
Race/ethnicity						<0.001
White	21,648 (85.0)	610 (94.6)	10,703 (94.4)	8403 (89.5)	1932 (47.3)	
African American	1366 (5.4)	13 (2.0)	164 (1.5)	333 (3.6)	856 (21.0)	
Hispanic	1340 (5.3)	14 (2.2)	256 (2.3)	347 (3.7)	723 (17.7)	
Asian/Pacific Islander	741 (2.9)	2 (0.3)	89 (0.8)	182 (1.9)	468 (11.5)	
Other/unknown	368 (1.5)	6 (0.9)	128 (1.1)	125 (1.3)	109 (2.7)	
Charlson-Deyo score						0.753
0	18,091 (71.1)	463 (71.8)	8077 (71.2)	6657 (70.9)	2894 (70.8)	
1	5492 (21.6)	140 (21.7)	2453 (21.6)	2027 (21.6)	872 (21.3)	
≥2	1880 (7.4)	42 (6.5)	810 (7.1)	706 (7.5)	322 (7.9)	
Histologic grade						<0.001
Well differentiated	913 (3.6)	29 (4.5)	434 (3.8)	358 (3.8)	92 (2.3)	
Moderately differentiated	8534 (33.5)	246 (38.1)	4216 (37.2)	3201 (34.1)	871 (21.3)	
Poorly differentiated	12,738 (50.0)	270 (41.9)	5055 (44.6)	4643 (49.5)	2770 (67.8)	
Undifferentiated	263 (1.0)	9 (1.4)	115 (1.0)	94 (1.0)	45 (1.1)	
Unknown	3015 (11.8)	91 (14.1)	1520 (13.4)	1094 (11.7)	310 (7.6)	
Tumor size						<0.001
<5 cm	10,617 (41.7)	235 (36.4)	4630 (40.8)	4238 (45.1)	1514 (37.0)	
>5–10 cm	3767 (14.79)	91 (14.1)	1648 (14.5)	1314 (14.0)	714 (17.5)	
>10–15 cm	304 (1.2)	8 (1.2)	104 (0.9)	104 (1.1)	88 (2.2)	
>15 cm	140 (0.6)	5 (0.8)	51 (0.5)	52 (0.6)	32 (0.8)	
Unknown	10,635 (41.8)	306 (47.4)	4907 (43.3)	3682 (39.20)	1740 (42.6)	
Days between preoperative therapy and surgery						<0.001
≤80 (quartile 1)	5536 (21.7)	147 (22.8)	2539 (22.4)	1992 (21.2)	858 (21.0)	
81–95 (quartile 2)	6985 (27.4)	178 (27.6)	3196 (28.2)	2656 (28.3)	955 (23.4)	
96–120 (quartile 3)	7641 (30.0)	187 (29.0)	3343 (29.5)	2784 (29.7)	1327 (32.5)	
>120 (quartile 4)	5301 (20.8)	133 (20.6)	2262 (20.0)	1958 (20.9)	948 (23.2)	
Clinical T category						<0.001
T0/T1/IS	1325 (5.2)	37 (5.7)	534 (4.7)	466 (5.0)	288 (7.1)	
T2	4535 (17.8)	142 (22.0)	2096 (18.5)	1555 (16.6)	742 (18.2)	
T3	16,003 (62.9)	399 (61.9)	7567 (66.7)	6109 (65.1)	1928 (47.2)	
T4	841 (3.3)	21 (3.3)	210 (1.9)	230 (2.5)	380 (9.3)	
Unknown	2759 (10.8)	46 (7.1)	933 (8.2)	1030 (11.0)	750 (18.4)	
Clinical N category						<0.001
N-negative	9634 (37.8)	230 (35.7)	4095 (36.1)	3444 (36.7)	1865 (45.6)	
N-positive	14,418 (56.6)	390 (60.5)	6746 (59.5)	5399 (57.5)	1883 (46.1)	
Unknown	1411 (5.5)	25 (3.9)	499 (4.4)	547 (5.8)	340 (8.3)	
Surgery type						<0.001
Partial esophagectomy	1671 (6.6)	90 (14.0)	1581 (13.9)	X	X	
Total esophagectomy	1296 (5.1)	84 (13.0)	1212 (10.7)	X	X	
Esophagectomy with laryngectomy and/or gastrectomy	8460 (33.2)	432 (67.0)	8028 (70.8)	X	X	
Esophagectomy, NOS	558 (2.2)	39 (6.0)	519 (4.6)	X	X	
Gastrectomy	3934 (15.4)	X	X	1810 (19.3)	2124 (52.0)	
Near-total or total gastrectomy	1589 (6.2)	X	X	643 (6.8)	946 (23.1)	
Gastrectomy with removal of portion of esophagus	6537 (25.7)	X	X	6088 (64.8)	449 (11.0)	
Gastrectomy with a resection in continuity with the resection of other organs	1333 (5.2)	X	X	780 (8.3)	553 (13.5)	
Gastrectomy, NOS	85 (0.3)	X	X	69 (0.7)	16 (0.4)	
Pathologic T category						<0.001
T0/IS	4485 (17.6)	165 (25.6)	2549 (22.5)	1474 (15.7)	297 (7.3)	
T1	4354 (17.1)	161 (25.0)	2212 (19.5)	1386 (14.8)	595 (14.6)	
T2	4839 (19.0)	111 (17.2)	2140 (18.9)	1859 (19.8)	729 (17.8)	
T3	10,575 (41.5)	197 (30.5)	4338 (38.3)	4432 (47.2)	1608 (39.3)	
T4	1210 (4.8)	11 (1.7)	101 (0.9)	239 (2.6)	859 (21.0)	
Pathologic N category						<0.001
N0	14,437 (56.7)	434 (67.3)	6992 (61.7)	5135 (54.7)	1876 (45.9)	
N1	6174 (24.3)	148 (23.0)	2869 (25.3)	2314 (24.6)	843 (20.6)	
N2	3079 (12.1)	50 (7.8)	1078 (9.5)	1274 (13.6)	677 (16.6)	
N3	1773 (7.0)	13 (2.0)	401 (3.5)	667 (7.1)	692 (16.9)	
Preoperative chemoradiotherapy						<0.001
No (chemotherapy only)	6501 (25.5)	48 (7.4)	872 (7.7)	1891 (20.1)	3690 (90.3)	
Yes	18,681 (73.4)	592 (91.8)	10,365 (91.4)	7394 (78.7)	330 (8.1)	
Unknown	281 (1.11)	5 (0.8)	103 (0.9)	105 (1.1)	68 (1.7)	

Abbreviations: IS, in situ; NOS, not otherwise specified.

**Table 2 cancers-15-05137-t002:** Three-year, five-year, and median overall survival (OS) by the gross ypTNM group among patients who underwent preoperative therapy for resectable esophageal and gastric adenocarcinoma (*n* = 25,463).

ypTNM Group		No.	3-Year OS, %	5-Year OS, %	Median OS, Years	*p*
Upper/middle esophageal adenocarcinoma (*n* = 645)						<0.001
N-negative	ypT1/2	344	67.2	54.5	6.1	
	ypT3/4	90	35.0	27.0	2.1	
N-positive	ypT1/2	93	36.1	28.8	2.2	
	ypT3/4	118	26.4	14.7	1.6	
Lower/abdominal esophageal adenocarcinoma (*n* = 11,340)						<0.001
N-negative	ypT1/2	5190	66.4	54.8	6.0	
	ypT3/4	1802	53.2	39.7	3.3	
N-positive	ypT1/2	1711	47.2	32.5	2.8	
	ypT3/4	2637	35.3	21.9	2.1	
Gastric cardia adenocarcinoma (*n* = 9390)						<0.001
N-negative	ypT1/2	3431	70.5	59.2	7.5	
	ypT3/4	1704	57.3	43.2	3.9	
N-positive	ypT1/2	1288	47.5	31.6	2.8	
	ypT3/4	2967	37.0	23.7	2.1	
Other stomach adenocarcinoma (*n* = 4088)						<0.001
N-negative	ypT1/2	1113	87.6	80.9	-	
	ypT3/4	763	69.0	56.8	7.1	
N-positive	ypT1/2	508	66.2	55.3	6.3	
	ypT3/4	1704	45.2	32.6	2.6	

**Table 3 cancers-15-05137-t003:** Cox proportional hazards regression model for overall survival among patients with lower or abdominal esophageal adenocarcinoma and those with gastric cardia adenocarcinoma (*n* = 17,876).

Variable	HR	95% CI	*p*
Tumor location (ref.: lower/abdominal esophagus)			<0.001
Gastric cardia	0.91	0.87–0.95	
Age (ref.: <50 years)			<0.001
50–59 years	1.15	1.07–1.25	
60–69 years	1.26	1.17–1.36	
≥70 years	1.62	1.50–1.75	
Sex (ref.: male)			<0.001
Female	0.87	0.82–0.93	
Charlson–Deyo score (ref.: 0)			<0.001
1	1.14	1.09–1.19	
≥2	1.26	1.17–1.35	
Histologic grade (ref.: well differentiated)			<0.001
Moderately differentiated	1.19	1.07–1.32	
Poorly differentiated	1.46	1.31–1.63	
Undifferentiated	1.47	1.21–1.8	
Margin status (ref.: negative)			
Positive	1.53	1.43–1.64	
Pathologic T category (ref.: T0/T1/IS)			<0.001
T2	1.20	1.13–1.27	
T3	1.42	1.35–1.49	
T4	1.93	1.67–2.22	
Pathologic N category (ref.: N0)			< 0.001
N1	1.54	1.47–1.61	
N2	2.01	1.89–2.13	
N3	2.50	2.30–2.72	
Preoperative therapy (ref.: chemotherapy)			<0.001
Chemoradiotherapy	1.31	1.24–1.39	

Abbreviations: CI, confidence interval; HR, hazard ratio; IS, in situ; ref., reference variable.

## Data Availability

The data that support the findings of this study are available from the corresponding author upon reasonable request.

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
