# Peer review of "Independent Stage Classification for Gastroesophageal Junction Adenocarcinoma"

_cancers, 2023, doi:10.3390/cancers15215137_

Round 1

Reviewer 1 Report

This manuscript entitled "Independent stage classification for gastroesophageal junction adenocarcinoma" provides a reasonable OS prediction nomogram, which includes preoperative therapy regimen, for patients with GEJ adenocarcinoma who underwent preoperative therapy and curative-intent surgery. The authors analyzed the highly reliable data acquired through credible statistical analysis of a large cohort. However, the authors should revise the following points.

Definition

The authors described that they investigated 25,463 patients with GEJ adenocarcinoma who received preoperative therapy followed by curative-intent surgery. However, they defined patients with GEJ adenocarcinoma as those who had adenocarcinoma of the lower third of the esophagus, abdominal esophagus, or gastric cardia, and thus the number of patients with GEJ adenocarcinoma was 11340. They should revise the description.

Chemoradiotherapy and chemotherapy

The authors showed that preoperative chemoradiotherapy was associated with lower 5-year OS rates than chemotherapy, even after adjustment for ypT and ypN categories. Previous RCTs comparing chemotherapy and chemoradiotherapy showed that chemoradiotherapy demonstrated no significant survival superiority over chemotherapy. However, these have consistently indicated that chemoradiotherapy promoted slightly more favorable survival, albeit without statistical significance. Please discuss why are these results different.

Prediction

The performance of the proposed model was not sufficient in terms of the ROC–AUC (0.669). I think that it is not acceptable. Do you have any suggestions for improvement?

Reviewer 2 Report

Thank you very much for giving me a good opportunity to review your article. Hirata et al. described “Independent stage classification for gastroesophageal junction adenocarcinoma” Independent new classification for GEJ adenocarcinoma is very attractive, however, this manuscript is not worthy of acceptance to “cancers” in its present form. A few more revisions are needed to be accepted.

Major comments:

1. It was clear that the prognosis was neatly divided into four groups according to ypT1/2 and ypT3/4 and the presence or absence of lymph node metastasis. Describe these with a Kaplan-Meier curve. Also compare the difference between the TNM stage and these four groups on the Kaplan-Meier curve.

2. The PRODIGY trial was conducted in Korea and reported the usefulness of DOS therapy for GEJ adenocarcinoma (J Clin Oncol 2021;39:2903-13.). Also, it was reported that a primary outcome was achieved in ASCO2023. Please add DOS as a NAC for GEJ adenocarcinoma.

3. Which technique is used in surgery for Siewert type 2 EGJ adenocarcinoma? The trans-hiatal approach or the trans-right thoracic approach? Please describe the technique for EGJ adenocarcinoma.
